# Case Report: Rare *IKZF1* Gene Fusions Identified in Neonate with Congenital *KMT2A*-Rearranged Acute Lymphoblastic Leukemia

**DOI:** 10.3390/genes14020264

**Published:** 2023-01-19

**Authors:** Laura N. Eadie, Jacqueline A. Rehn, James Breen, Michael P. Osborn, Sophie Jessop, Charlotte E. J. Downes, Susan L. Heatley, Barbara J. McClure, David T. Yeung, Tamas Revesz, Benjamin Saxon, Deborah L. White

**Affiliations:** 1Blood Cancer Program, Precision Cancer Medicine Theme, South Australian Health & Medical Research Institute, Adelaide, SA 5000, Australia; 2Faculty of Health and Medical Sciences, University of Adelaide, Adelaide, SA 5000, Australia; 3South Australian Genomics Centre (SAGC), Adelaide, SA 5000, Australia; 4Robinson Research Institute, University of Adelaide, Adelaide, SA 5006, Australia; 5Australian & New Zealand Children’s Haematology/Oncology Group, Clayton, VIC 3168, Australia; 6Australasian Leukaemia & Lymphoma Group, Richmond, VIC 3121, Australia; 7Department of Haematology & Oncology, Women’s & Children’s Hospital, Adelaide, SA 5000, Australia; 8Royal Adelaide Hospital, Adelaide, SA 5000, Australia; 9Faculty of Sciences, Engineering and Technology, University of Adelaide, Adelaide, SA 5000, Australia; 10Clinical Services and Research Division, Australian Red Cross Blood Service, Adelaide, SA 5000, Australia; 11Australian Genomics Health Alliance, Parkville, VIC 3052, Australia

**Keywords:** congenital acute lymphoblastic leukemia, infant ALL, *KMT2A*-rearranged ALL, case report, chromosomal abnormalities, fusion gene, *IKZF1* translocation, mRNA-sequencing

## Abstract

Chromosomal rearrangements involving the *KMT2A* gene occur frequently in acute lymphoblastic leukaemia (ALL). *KMT2A*-rearranged ALL (*KMT2Ar* ALL) has poor long-term survival rates and is the most common ALL subtype in infants less than 1 year of age. *KMT2Ar* ALL frequently occurs with additional chromosomal abnormalities including disruption of the *IKZF1* gene, usually by exon deletion. Typically, *KMT2Ar* ALL in infants is accompanied by a limited number of cooperative le-sions. Here we report a case of aggressive infant *KMT2Ar* ALL harbouring additional rare *IKZF1* gene fusions. Comprehensive genomic and transcriptomic analyses were performed on sequential samples. This report highlights the genomic complexity of this particular disease and describes the novel gene fusions *IKZF1::TUT1* and *KDM2A::IKZF1*.

## 1. Introduction

The histone-lysine [K] Methyl Transferase 2A (*KMT2A*) on chromosome 11q23 is a pathogenic driver gene in acute lymphoblastic leukemia (ALL). *KMT2A*-rearranged ALL (*KMT2A*r ALL) has poor long-term survival rates and is the most common ALL subtype in infants (<1 year of age) [1], comprising >70% of all ALL diagnoses in this age group [2]. It is often associated with hyperleukocytosis and central nervous system (CNS) involvement [3]. While *KMT2A*r ALL frequently occurs with additional chromosomal abnormalities including disruption of the *IKZF1* gene (chromosome 7p12), usually by exon deletion [4], *KMT2A*r ALL in infants typically presents with few co-occurring alterations [5]. However, we report a genomically complex case of aggressive congenital *KMT2A*r ALL harboring additional rare and novel *IKZF1* gene fusions.

## 2. Case Report

A full-term neonate was delivered to a mother with an unremarkable antenatal history. At three days old the infant had a complete blood count performed in the setting of hyperbilirubinemia and poor feeding. This revealed a leukocytosis of 136 × 10^9^/L (98% lymphoblasts), with normal red cell and platelet parameters (hemoglobin 171 × 10^9^/L, platelets 183 × 10^9^/L). Immunophenotyping of peripheral blood mononuclear cells (PBMNC) confirmed B-ALL (70% blasts; 93% CD19/CD34^+^, 39% CD10/CD19/CD34^+^). Cytogenetics demonstrated a 47XX karyotype with trisomy 8 and translocations t(4;11)(q21;q23) and t(7;11)(q11.2;p11.2). The *KMT2A::AFF1* (*MLL::MLLT2/AF4*) *KMT2A* rearrangement was confirmed by fluorescence in situ hybridization. No diagnostic bone marrow biopsy was performed. A lumbar puncture revealed CNS involvement.

The infant was treated with induction chemotherapy as per the Interfant-06 protocol [6] and achieved a minimal residual disease (MRD) level of 5 × 10^−3^ by the end of induction using Allele Specific Oligonucleotide PCR for *IgH* rearrangement. However, further intensive chemotherapy was delayed by significant toxicities, including vincristine-related polyneuropathy requiring mechanical ventilation. Following two weeks of dose-reduced mercaptopurine and oral methotrexate maintenance, azacitidine was administered (2.5 mg/kg for 5 days) [7] with the intention of bridging to hematopoietic stem cell transplant if remission was obtained. Unfortunately, the bone marrow blast percentage rose to 5% at the end of this cycle, and despite further azacitidine and consolidation chemotherapy as per the protocol AALL15P1 [7], by the next bone marrow examination, lymphoblasts had risen to 55%. Subsequent blinatumomab combined with intrathecal chemotherapy resulted in a morphological remission with MRD of 10^−2^. However, the second blinatumomab course was complicated by the development of facial nerve palsy secondary to an extradural leukemic chloroma impinging on her facial nerve. This heralded florid morphological relapse with 42% bone marrow blasts, which exhibited complete loss of CD19 on flow cytometry (93% CD19/34^+^ at diagnosis). The patient failed to respond to either FLAG-IDA or inotuzumab and succumbed at nine months of age. The timeline of treatments and response assessments are summarized in Figure 1, and immunophenotyping flow cytometric dot plots are shown in Appendix A.

## 3. Results

Transcriptomic sequencing (mRNA-Seq) performed on PBMNCs at diagnosis identified the *KMT2A::AFF1* gene fusion, with low number of reads. Two *IKZF1* gene fusions were also identified: *IKZF1::TUT1* and *KDM2A::IKZF1* (Figure 2A, Appendix A). Both *IKZF1* fusions were validated by PCR and Sanger sequencing (Figure 2B). Significantly, this is the first time the *KDM2A::IKZF1* and *IKZF1::TUT1* gene fusions have been described, and the first report of *KMT2A*r ALL with co-occurring *IKZF1* fusions [8]. Although *KDM2A* and *TUT1* are both in the same karyotypic region of chromosome 11 (11q13.2 and 11q12.3, respectively), they are separated by >190 genes, with a genetic distance corresponding to a recombination frequency of >3.8% [9]. Thus, these two fusions likely represent separate genomic events. Multiplex ligation-dependent probe amplification (MLPA) is a PCR-based method for quantification of DNA copy numbers and a reliable method for copy number variation (CNV) genotyping. We used two different MLPA probe mixes (P202 and P335, MRC Holland) to determine CNV in genomic DNA. No deletions or duplications were detected in any of the genes assayed; of importance, no deletions were detected in *IKZF1* exons 1–8 (Appendix A).

The *KMT2A::AFF1* gene fusion observed here is the most common fusion observed in infant disease (49% of all infant *KMT2A*r leukemias [1]). However, the breakpoint in *KMT2A* exon 9 (Figure 3A) is rarely observed in infant *KMT2A::AFF1* leukemia (19%, compared with the frequently observed exon 11 breakpoint) [8]. Upon formation of *KMT2A::AFF1*, the entire C-terminal portion of *KMT2A* is lost; this region contains domains important for post-translational regulation and mediation of protein–protein interactions. The lost regulatory and H3K4 methyltransferase activity lead to the widespread epigenetic dysregulation observed in *KMT2A*r patients [1]. The *KMT2A* portion retained in the fusion harbors binding motifs for proteins, such as menin and LEDGF, which are critical for leukemic transformation [10,11], and domains to facilitate KMT2A’s DNA-binding capacity. The *AFF1* portion of the fusion lacks a degron sequence, likely perturbing the protein’s degradation rate. This is supported by high *AFF1* expression in this patient, as compared with *AFF1* expression in all other B-ALL samples in our patient cohort. Conversely, while *KMT2A* gene expression was not increased, elevated expression of the homeobox gene *MEIS1* was observed as is typical of *KMT2A*r patients [12] (Figure 3B). It should be noted that, in an in vivo model, *KMT2A::AFF1* fusions were incapable of inducing leukemia in isolation, suggesting additional genomic aberrations are required [13].

The Terminal Uridylyl Transferase 1 (*TUT1*) gene has previously been reported as a fusion partner in T-cell lymphoblastic lymphoma patients [15]; however, *IKZF1::TUT1* is a novel fusion gene. *TUT1* encodes a nucleotidyl transferase enzyme that may play a role in controlling gene expression and cell proliferation; however, its role in oncogenesis remains unclear. The novel *IKZF1::TUT1* fusion is the predominant gene fusion in this patient at all disease timepoints (Figure 2A, Appendix A). Interestingly, only a small portion of the *IKZF1* gene, containing no functional domains, is present. Thus, it is likely that the *TUT1* fusion partner is driving the putative leukemic activity of this fusion. Additionally, the diagnosis sample exhibits the highest expression level of the *TUT1* gene observed in our patient cohort (Figure 3B). Limited functional data for *TUT1* gene fusions exist; however, given that the RNA recognition motif is truncated while the nuclear localization sequence remains (Figure 3C), it is possible that aberrant gene expression occurs as a result of *IKZF1::TUT1*.

The second novel fusion involving *IKZF1*, *KDM2A::IKZF1*, incorporates the catalytic domain of *KDM2A* and all *IKZF1* functional domains (Figure 3D). The encoded Ikaros protein is a transcription factor with key regulatory functions in lymphopoiesis [16]. *IKZF1* is a leukemic driver and functions as a tumor suppressor and loss of Ikaros function, either by mutation or deletion, is frequently observed in B-cell ALL [17] as well as other hematological malignancies [18]. Ikaros loss of function alterations are associated with poor prognosis and inferior treatment outcomes [19,20,21]. However, currently no outcome data for *IKZF1* gene fusions are available, most likely due to the rarity of these alterations (Table 1). Whether these fusions are driver alterations and contribute to leukemic development also remains to be determined.

Transcriptomic sequencing was performed on mesenchymal stem cells generated from hair follicles, representing a germline sample, as well as sequential samples taken when the patient was refractory following induction therapy and while undergoing blinatumomab therapy prior to relapse (Figure 2A). Interestingly, the *KDM2A::IKZF1* and *KMT2A::AFF1* gene fusions were no longer detectable by mRNA-Seq following blinatumomab therapy, suggesting the *IKZF1::TUT1* gene fusion may be responsible for driving relapse. 

Infant *KMT2A*r ALL cases normally present with pro-B-cell blasts with the immunophenotype: B220/CD43/19/34/22/TdT/CyCD79a^+^, CD10/BPI/IgM^–^ [22,23]. However, immunophenotypic analyses of PBMNCs from this infant detected an atypical immunophenotype at diagnosis with a clear population of CD10^+^ cells (93% CD19/CD34^+^, 39% CD10/CD19/CD34^+^) suggesting the presence of two leukemic populations (Figure 1 and Appendix A). Similarly, at both the refractory (13% CD19/CD34^+^, 0.6% CD10/CD19/CD34^+^) and on-blinatumomab timepoints (CD19 negative, 2.8% CD10/CD34^+^), two immunophenotypic populations were present. Comparing these data with the gene fusions identified by transcriptomic sequencing (Figure 2A), it is likely that the CD19^-^ clone present during blinatumomab therapy harbored the *IKZF1::TUT1* gene fusion, while the CD10/CD19^+^ clone harbored the *KDM2A::IKZF1* and *KMT2A::AFF1* gene fusions. However, this would require confirmatory sequencing of each cell population.

**Table 1 genes-14-00264-t001:** *IKZF1* gene fusions previously reported in acute lymphoblastic leukemia.

Fusion	Chromosome Abnormality	Disease	Number of Cases	Description	*IKZF1* Exon Retention	Predicted Function	Ref
*DNAH14::IKZF1*	t(1;7)(q42;p12)	B-ALL	1	in-frame; exon 36 to exon 4 ZNF2-4 domains (DNA-binding function) and ZNF5-6 domains containing dimerization sites are truncated	exons 1–4	Similar to the IK6 isoform, loss-of-function allele	[24]
*ETV6::IKZF1*	t(7;12)(p12;p13)	B-ALL	1	out-of-frame; intron 2 to intron 3 No functional *IKZF1* domains are present	exons 1–3	Likely abolishes the function of Ikaros protein	[18]
*FIGNL1::IKZF1*	t(7;7)(p12;p12)	B-ALL	1	out-of-frame; exon 4 to exon 4* Majority of functional *IKZF1* domains are retained; ZNF1 domain (DNA-binding function) is truncated	exons 5–8	Altered transcriptional regulation	[25]
*IKZF1::CDK2*	t(7;12)(p12;q13)	B-ALL	1	out-of-frame; exon 3, no *CDK2* breakpoint details provided No functional Ikaros domains are present	exons 1–3	Likely abolishes the function of Ikaros protein	[26]
*IKZF1::ETV6*	t(7;12)(p12;p13)	B-ALL	1	out-of-frame; intron 3 to intron 2 No functional *IKZF1* domains are present	exons 1–3	Likely abolishes the function of Ikaros protein	[18]
*IKZF1::FIGNL1*	t(7;7)(p12;p12)	B-ALL	2	out-of-frame; intron 3 to 5′ UTR exon 2 and exon 3 to 13691 bp downstream No functional *IKZF1* domains are present	exons 1–3	Likely abolish the function of Ikaros protein	[18,27]
*IKZF1::NUTM1*	t(7;15)(p12;q14)	B-ALL	1	in-frame; exon 7 to exon 2 Some functional *IKZF1* domains are retained; ZNF5-6 domains containing dimerization sites are truncated	exons 1–7	Altered transcriptional regulation	[26]
*IKZF1::SETD5*	t(3;7)(p25;p12)	B-ALL	1 ^#^	in-frame; exon 3, no *SETD5* breakpoint details provided No functional *IKZF1* domains are present	exons 1–3	Likely abolishes the function of Ikaros protein	[26]
*IKZF1::TRPV2*	t(7;17)(p12;p11)	B-ALL	1	out-of-frame; no exon/intron details provided No functional *IKZF1* domains are present	breakpoint not specified	Likely abolishes the function of Ikaros protein	[26]
*IKZF1::ZEB2*	t(2;7)(q22;p12)	B-ALL	1	in-frame; exon 3 to exon 5 No functional *IKZF1* domains are present	exons 1–3	Likely abolishes the function of Ikaros protein	[18]
*SETD5::IKZF1*	t(3;7)(p25;p12)	B-ALL	1 ^#^	in-frame; no *SETD5* breakpoint details provided, exon 4 All functional *IKZF1* domains are retained	exons 4–8	Altered transcriptional regulation	[26]
*STIM2::IKZF1*	t(4;7)(p15;p12)	B-ALL	1	No fusions details provided	breakpoint not specified	No fusion details provided	[28]
*IKZF1::ABCA13*	t(7;7)(p12;p12)	T-ALL	1	out-of-frame; exon to intron Some functional *IKZF1* domains are retained; ZNF4 domain (DNA-binding function) and ZNF5-6 domains containing dimerization sites are truncated	exons 1–5	Reduced expression of *IKZF1*; likely abolishes the function of Ikaros protein	[29]
*IKZF1::NOTCH1*	t(7;9)(p12;q34)	T-ALL	1	in-frame; exon to exon Some functional *IKZF1* domains are retained; ZNF4 domain (DNA-binding function) and ZNF5-6 domains containing dimerization sites are truncated	exons 1–5	Altered transcriptional regulation	[30]
*IKZF1::NOTCH1*	t(7;9)(p12;q34)	T-ALL	1 ^$^	out-of-frame; intron to intron Some functional *IKZF1* domains are retained; ZNF4 domain (DNA-binding function) and ZNF5-6 domains containing dimerization sites are truncated	exons 1–5	Altered transcriptional regulation	[27]
*NOTCH1::IKZF1*	t(7;9)(p12;q34)	T-ALL	1 ^$^	out-of-frame; exon to intron Some functional *IKZF1* domains are retained; ZNF1-4 domains containing dimerization sites are truncated	exons 6–8	Altered transcriptional regulation	[27]

Note 1: Retained *IKZF1* exons refers to *IKZF1* transcript variant 1, NM_006060 except for (*), which relates to alternate transcript variant 15, NM_001291845. Note 2: # *IKZF1::SETD5* and *SETD5::IKZF1* fusions were observed in the same patient. Note 3: $ *IKZF1::NOTCH1* and *NOTCH1::IKZF1* fusions were observed in the same patient.

## 4. Discussion

Here, we describe a rare case of congenital *KMT2A*r ALL presenting with co-occurring *IKZF1* gene fusions and a predictably aggressive disease trajectory. We report for the first time the novel *IKZF1::TUT1* and *KDM2A::IKZF1* gene fusions. Rearrangements involving *KMT2A* are commonly retained in relapsed infant ALL [5,31,32]; however, in this case, the *KMT2A::AFF1* gene fusion did not appear to be the lesion driving leukemic relapse. Instead, our data suggest that relapse was driven by *IKZF1::TUT1*. This gene fusion remained in all samples investigated, including the on-blinatumomab therapy sample taken immediately prior to relapse. Conversely, the *KMT2A::AFF1* gene fusion was only detected in the diagnosis and refractory post-induction samples, highlighting a key role for *IKZF1::TUT1* in disease pathogenesis. Intriguingly, both *IKZF1* gene fusions are predicted to be out-of-frame (Appendix A); however, our data demonstrate the *IKZF1* gene is still expressed (Figure 3B). This is not unprecedented, and it has previously been observed that out-of-frame fusions can cause transcriptional activation/repression of genes involved in the fusions leading to increases or decreases of their expression and the associated functional outcomes [29].

Ikaros is a lymphoid transcription factor with a tumor-suppressive function. Alterations that knock out the Ikaros function, such as the gene fusions described here, would presumably also affect Ikaros target genes including signal transducers (c-kit, Flt3, Il7r), pre-B-cell receptor signaling proteins (Syk) and cell cycle regulators (Cdkn2a, Cdkn1a) [33]. Indeed, altered Flt3 [34] and Syk [35] expression has been reported in *KMT2A*r ALL. Studies have demonstrated Flt3 inhibitors are active against *KMT2A*r disease in vivo [36] and, when administered in combination with various chemotherapeutics including some of those used here (dexamethasone, cytarabine, asparaginase), effectively kill *KMT2A*r cells in vitro [37,38]. More recently, a Children’s Oncology Group trial has demonstrated the benefit of Flt3 inhibitor lestaurtinib in combination with chemotherapy (Interfant 99-based induction regimen) for treating infants with *KMT2A*r ALL [39]. Similarly, the combination of vincristine and the Syk inhibitor entospletinib demonstrated enhanced efficacy in in vivo models of infant *KMT2A*r ALL compared with either agent alone [40]. However, entospletinib as a treatment for ALL has yet to enter clinical trials. A retrospective case study of 11 infants with *KMT2A*r ALL demonstrated the efficacy of blinatumomab in patients with relapsed/refractory disease [41], and pre-clinical efficacy was observed in recent in vivo models assessing azacitidine in combination with venetoclax [42]. However, we observed poor response to the Interfant-06 induction and Children’s Oncology Group consolidation protocols (AALL15P1), which comprise various chemotherapy agents, including those detailed above (Figure 1), and while azacitidine was well tolerated, the patient soon became resistant. The immunotherapies blinatumomab and inotuzomab [43] as well as the FLAG-IDA relapse regimen also failed. Case reports such as this highlight the urgent need for new targeted therapies to improve outcome in *KMT2A*r infant ALL. We also demonstrate the importance of gene sequencing to comprehensively dissect the underlying genomic complexity of a disease like ALL and identify co-occurring alterations that may impact treatment outcomes.

## Figures and Tables

**Figure 1 genes-14-00264-f001:**
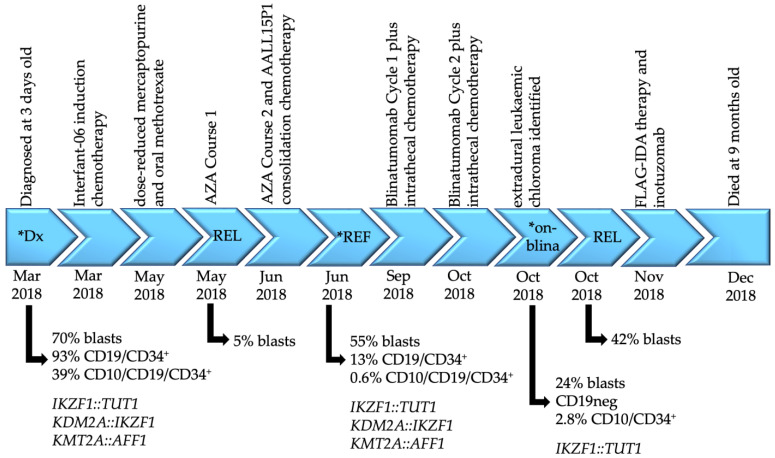
Disease timeline of patient CHI_0391. Disease timepoints, associated treatments, blast percentages with the corresponding immunophenotype where available and gene fusions identified by mRNA-Seq are indicated. Asterisks denote the samples analyzed by transcriptomic sequencing in this report. Interfant-06 induction chemotherapy includes prednisone, dexamethasone, vincristine, cytarabine, daunorubicin, PEG-asparaginase, methotrexate, bortezomib and melphalan; AALL15P1 consolidation protocol includes cyclophosphamide, mercaptopurine, cytarabine, methotrexate, hydrocortisone and methotrexate. Dx = diagnosis; REF = refractory; REL = relapse; AZA = azacytidine; blina = blinatumomab; FLAG-IDA = fludarabine, cytarabine, granulocyte-colony stimulating factor, idarubicin.

**Figure 2 genes-14-00264-f002:**
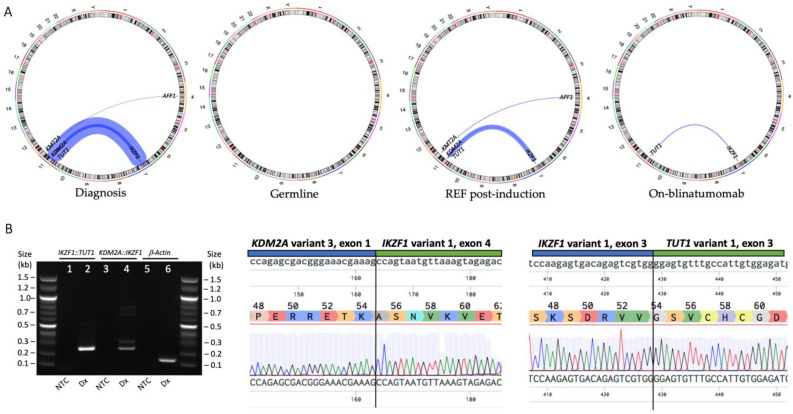
(**A**) Circos plot representation of gene fusions identified by transcriptomic sequencing of PBMNC at diagnosis, MSC representing a germline sample, BMMNC from a refractory sample after commencing induction therapy and BMMNC from a sample while undergoing blinatumomab therapy. (**B**) The *IKZF1::TUT1* and *KDM2A::IKZF1* gene fusions were detected by breakpoint RT-PCR in the diagnosis PBMNC sample. *IKZF1::TUT1* product size = 261 bp, *KDM2A::IKZF1* product size = 256 bp, *β-Actin* = 193 bp. The fusions were validated by Sanger sequencing. The fusion breakpoints are shown in the upper panels and delineated with a vertical line, the amino acid reference sequences in the middle panels and the sequencing trace in the lower panels. Sequencing was aligned with Benchling software. Abbreviations: PBMNC = peripheral blood mononuclear cells; MSC = mesenchymal stem cells; BMMNC = bone marrow mononuclear cells; REF = refractory; Dx = diagnosis; NTC = no template control.

**Figure 3 genes-14-00264-f003:**
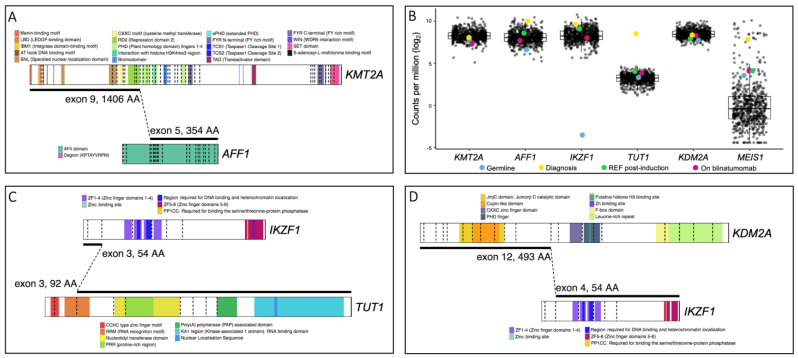
Schematic representations of the *KMT2A::AFF1*, *IKZF1::TUT1* and *KDM2A::IKZF1* gene fusions were created using ProteinPaint [14]. (**A**) The *KMT2A::AFF1* gene fusion retains DNA- and protein-binding domains, nuclear localization domains and the repression domain responsible for HDAC1 and HDAC2 recruitment. The menin- and LEDGF-binding domains are critical for leukemic transformation involving KMT2A, as both menin and LEDGF are essential oncogenic co-factors for KMT2A. The interaction of the three proteins is necessary for leukemic transformation [10,11]. (**B**) Boxplots of differential expression levels of five genes involved in gene fusions identified in patient CHI_0391. Gene expression levels in germline, diagnosis, refractory post-induction therapy and on-blinatumomab treatment samples from patient CHI_0391 (colored dots) are denoted and compared with samples from our B-ALL patient cohort (black dots; 592 patient samples at various disease timepoints including the 4 highlighted samples from CHI_0391. See Appendix A for cohort characteristics). The data are expressed as log normalized counts per million. (**C**) The *IKZF1::TUT1* gene fusion retains a small portion of *IKZF1* but the majority of *TUT1,* encompassing the nuclear localization sequence and truncating the RNA recognition motif. (**D**) The *KDM2A::IKZF1* gene fusion retains the catalytic domain of *KDM2A* and all functional domains of *IKZF1*.

## Data Availability

The data presented in the study are deposited in the European Genome Phenome Archive.

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
