# Peer review of "Case Report: Rare IKZF1 Gene Fusions Identified in Neonate with Congenital KMT2A-Rearranged Acute Lymphoblastic Leukemia"

_genes, 2023, doi:10.3390/genes14020264_

Round 1
Reviewer 1 Report
The article 'Case Report: Rare IKZF1 gene fusions identified in neonate with congenital KMT2A-rearranged Acute Lymphoblastic Leukemia' by Laura N Eadie et al.
The article is structured well, and the work is relevant for the readership of ‘Genes.’
There are the following comments towards the current version of the manuscript.
1.Figure1 A, the author did Immunophenotyping of peripheral blood mononuclear cells of (PBMNC) by flow cytometry, the author should show the picture of flow cytometry, which is more convincing.
2.Figure2B, KDM2A::IKZF1 product size=256 bp in Dx, but for this 4-lane band, in addition to the 256 bp band, there is also a faint band about 300 bp, How does the author understand this band?
3.For figure 2A, does the thickness of the blue line represent the number of mRNA-Seq reads of this gene fusion? That means that in this diagnosis, the IKZF1::TUT1 gene reads are higher than the KMT2A::AFF1 gene reads.
After blinatumomab therapy, KMT2A::AFF1 was not detected, but IKZF1::TUT1 was detected. As the author said, this suggests that KZF1::TUT1 gene fusion may be responsible for driving relapse.
But IKZF1::TUT1 is also reduced after blinatumomab therapy, which suggests that KZF1::TUT1 gene fusion may not be responsible for driving relapse.
4.For figure 2A, the name of gene fusion in blue line is too small.Please change the color and size so that readers can see it
5.For figure 3B, The color of Germline and On blinatumomab overlaps with the black dots and cannot be seen. Please change the color.
6. Some references are in the wrong format with hyperlinks, please correct it.
Reviewer 2 Report
Eadie et al apply RNAseq to identify two novel IKZF1 fusions accompanying a congenital KMT2A positive ALL case. They discuss the clinical course of the patient focusing on molecular the events and review IKZF1 fusions previously reported in acute lymphoblastic leukemia.
The manuscript is overall well written, results are well structured and presented and the literature has been thoroughly reviewed.
A couple of points could be improved:
- The authors quantify the expression levels of the rearranged genes and compare it with a B-ALL cohort. It would be nice to have some information about the basic characteristics of the cohort, especially if other fusions involving the same genes are found in other patients.
- Figure 1 (disease timeline) could include the molecular events the same way that immunophenotype has been included, it would be visually helpful.
